# Analysis of the Results of the Borowiec SLR Station (7811) for the Period 1993–2019 as an Example of the Quality Assessment of Satellite Laser Ranging Stations

**DOI:** 10.3390/s22020616

**Published:** 2022-01-13

**Authors:** Stanisław Schillak, Paweł Lejba, Piotr Michałek, Tomasz Suchodolski, Adrian Smagło, Stanisław Zapaśnik

**Affiliations:** Centrum Badań Kosmicznych Polskiej Akademii Nauk (CBK PAN), Bartycka 18A, Obserwatorium Astrogeodynamiczne CBK PAN Borówiec, 00-716 Warszawa, Poland; plejba@cbk.poznan.pl (P.L.); misiek@cbk.poznan.pl (P.M.); suchodolski@cbk.poznan.pl (T.S.); amaglo@cbk.poznan.pl (A.S.); s.zapasnik@cbk.poznan.pl (S.Z.)

**Keywords:** satellite geodesy, satellite laser ranging (SLR), satellite orbits, station coordinates, Borowiec SLR, LAGEOS satellites

## Abstract

This paper presents the results of an orbital analysis of satellite laser ranging data performed by the Borowiec SLR station (7811) in the period from July 1993 to December 2019, including the determination of the station positions and velocity. The analysis was performed using the GEODYN-II orbital program for the independent monthly orbital arcs from the results of the LAGEOS-1 and LAGEOS-2 satellites. Each arc was created from the results of the laser observations of a dozen or so selected stations, which were characterized by a large number of normal points and a good quality of observations. The geocentric and topocentric coordinates of the station were analyzed. Factors influencing the uncertainty of the measurements were determined: the number of the normal points, the dispersion of the normal points in relation to the orbits, and the long-term stability of the systematic deviations. The position leap at the end of 2002 and its interpretation in ITRF2014 were analyzed. The 3D stability of the determined positions throughout the period of study was equal to 12.7 mm, with the uncertainty of determination being at the level of 4.3 mm. A very high compliance of the computed velocity of the Borowiec SLR station (24.9 mm/year) with ITRF2014 (25.0 mm/year) was found.

## 1. Introduction

The elaboration and control of the results of SLR observations are presented in several commonly available datasets. International Laser Ranging Service (ILRS) analyses are most often used, including Quarterly/Monthly Global Performance Card [1]; Analysis Centers: ILRS ASC Product and Information Server [2]; Hitotsubashi University: Multi-Satellite Bias Analysis Report [3]; Joint Center for Earth Systems Technology (JCET); DGFI-TUM ILRS Analysis Centre [4]; and SLR Observatory Zimmerwald: ILRS Combined Range bias Report [5]. All these centers have advantages and disadvantages, but generally do not contain complete information about the activity of a given station, especially over a long period. The JCET center, which enables the graphical presentation of the results over any period, is very useful. The Hitotsubashi University center is the best for the ongoing control of the results; after a few hours, it is possible to verify the results for several satellites.

The aim of this work is fully present the results of observations of one SLR station over a long period for the example of the Borowiec SLR station (7811). A similar study has already been carried out by the first author for the Riyadh SLR station, summing up several years of the operation of this station [6].

The satellite laser ranging station at Borowiec started operating as a second-generation station in May 1988, sending its first observation results to the international data center EUROLAS Data Center (EDC) and NASA’s Crustal Dynamics Data Information System (CDDIS NASA). In 1991, a new third-generation laser was installed, which allowed for a significant improvement in the quality and quantity of results [7]. The system operated at a good quality level from July 1993, after an error in the frequency of the time interval counter was corrected. From then until March 2010, satellite observations were conducted without significant interruptions. In March 2010, due to the wear of the laser, after almost 20 years of operation the observations were stopped. The cause of this was the destruction of the mirrors in both heads, with new heads no longer being available. The further operation of this station could only be ensured by the purchase of a new laser. Efforts were successfully completed in 2013. The Lithuanian EKSPLA PL-2250 laser was purchased and fully met expectations. In 2014, works related to the installation of the laser and the modernization of the system were carried out, including the replacement of the main and secondary mirrors of the tracking telescope, as well as the replacement of the tables and mirrors of the Coude system. The first observations after a five-year break were obtained in March 2015 [8]. Since then, observations have been carried out without interruptions until now (2021).

The presented work contains an analysis of the results obtained in the frame of the International Laser Ranging Service (ILRS) [9] by the station at Borowiec for the period from July 1993 to December 2019. The independent geocentric and topocentric coordinates of the stations in monthly periods, together with the standard deviations of their determination, were computed using the NASA Goddard Space Flight Center GEODYN-II orbital program [10] for both the LAGEOS-1 and LAGEOS-2 satellites. The orbital RMS of the fit, range bias, and long-term stability were determined separately for each satellite. The RMS (coordinates stability) of the designated coordinates and the mean deviation of the coordinates from the International Terrestrial Reference Frame (ITRF2014) [11] were also obtained. The station velocity and azimuth of the station’s movement were determined. The results are presented in graphs illustrating changes in the position of the station over the 27 years studied.

The precise determination of the position and velocity of the station requires observation periods longer than several years. Based on the long time series of the SLR station, the laser technique enables the determination of absolute positions stability on the Earth’s surface. The task of this work is the orbital verification of the quantity and quality of observations of the Borowiec SLR station over the almost thirty years of the station’s operation. The choice of the LAGEOS satellites used to determine the coordinates of the stations is justified by several factors that significantly reduce disturbances in their orbital motion: the low influence of the Earth’s gravitational field, harmonics up to the degree and order of 20, the lack of atmospheric drag, and the lower influence of the Earth’s albedo. In the case of low geodetic satellites (LEO) (Ajisai, Starlette, Stella, Larets, and LARES) there is a very strong influence of the Earth’s gravitational field up to harmonics of the degree and order of 100 × 100. These are much harder to accurately determine. A very significant influence on these satellites is atmospheric drag, which is difficult to determine. The worse configuration of their orbits is also important, limiting the number of observations that can be made simultaneously by several neighboring stations. The multi-satellites method [12] based on LAGEOS-1, LAGEOS-2, LARES, Low Earth Orbit (LEO) satellites, and Global Navigation Satellite Systems (GNSS) satellites is taken into account for the determination the coordinates of these stations, but the results have not yet been fully verified.

The aim of this work is to answer several important questions regarding the operation of these station. How did the positions of these stations change and what were the reasons for these changes? Were the velocity and azimuth of the station movement maintained at fixed levels? What was the quality of the determined coordinates, and was it fixed or subject to changes? What were the reasons behind the changes in the quality of the observations? Were the results consistent for the LAGEOS-1 and LAGEOS-2 satellites? Were the obtained results consistent with the results of other analysis centers? What actions should be taken to significantly improve the quality and quantity of observations in the near future?

## 2. Materials and Methods-Hardware of Borowiec SLR Station in 1993–2019

The measurements of the LAGEOS satellites analyzed in this paper were performed by the Borowiec SLR system, which has been continuously developed since 1988. Over the years 1993–2019, many significant improvements in the system have been made concerning the receiving telescope, laser module, receiver system, time and frequency standards, software, control computers, and meteorological instrumentation. All these changes, modifications, and upgrades have contributed to the visible increase in the effectiveness of the laser system, including the tracking of LEO, MEO satellites, and LEO space debris, as well as to the significant growth of the RMS of the laser system measurements from 40 mm in 1993 to 18 mm in 2019 [7,8,13,14]. The most important changes and upgrades of the Borowiec station are listed in Table 1.

Table 2 shows a comparison of the Borowiec SLR station in 1993 and 2019. The table contains the most important information regarding the laser, telescope, receiver system, time base, computer, calibration, and meteo instrument. The number of photoelectrons (*n_ph_)* generated by the detector can be estimated by the radar link equation in the form [15]:(1)$$n_{ph}=\eta_{q}\left(E_{T}\frac{\lambda }{hc}\right)\eta_{t}G_{t}\sigma {\left(\frac{1}{4\pi R^{2}}\right)}^{2}A_{r}\eta_{r}T_{a}^{2}T_{c}^{2}$$
where $\eta_{q}$ is the detector quantum efficiency, $E_{T}$ is the laser pulse energy, λ is the laser wavelength, h is Planck’s constant, c is the speed of light in a vacuum, $\eta_{t}$ is the transmit optics efficiency, $G_{t}$ is the transmitter gain (inversely proportional to the square of laser beam divergence θ), σ is the satellite/space debris optical cross-section, R is the range to the target, $A_{r}$ is the effective area of the telescope receiver aperture, $\eta_{r}$ is the efficiency of the receiver optics, $T_{a}$ is the one-way atmospheric transmission, and $T_{c}$ is the one-way transmissivity of cirrus clouds.

Assuming the range of LAGEOS to be 5950 km (elevation 90°), $T_{a}$ to be 0.8, $T_{c}$ to be 1.0 for an elevation of 90° (taken from [15]), σ to be 7 × 10^6^ m^2^ for LAGEOS [16], the transmit optics efficiency to be 0.7, and the efficiency of the receiver optics to be 0.7, the number of photoelectrons received in a one-second cycle is 2.08 and 4.16 for the laser system in 1993 and 2019, respectively. In 1993, the Borowiec station tracked seven objects with 350 confirmed passes. In 2019, it tracked 88 objects from the LEO and MEO regimes, including space debris, with 1468 good passes recorded.

## 3. Data and Methods-Orbits Determination of LAGEOS Satellites

The NASA GSFC GEODYN-II orbital program [10] was used to determine the orbits of the LAGEOS-1 and LAGEOS-2 satellites. The determination of the coordinates of the stations was carried out using the method presented in [17]. In this method, the geocentric coordinates are computed for one station from each arc and the coordinates of the other stations are fixed in ITRF2014. The method is based on two steps: in the first one, the LAGEOS orbits are computed from the core station data; in the second, the geocentric positions of one station are computed from the differences between the satellite coordinates and the “a priori” ITRF2014 station position for all normal points of a given station per arc. This method enables a good control of the position of the station in relation to the fixed positions of all stations defined in ITRF2014. As a result, all small deviations due to orbit inaccuracy or ITRF2014 will be detected in the station coordinates and thus determined.

The orbits of the LAGEOS-1 and LAGEOS-2 satellites were computed from the results of 34 selected SLR stations in the form of 120-second normal points [18] taken from the EUROLAS Data Center (EDC). The parameters and models used in the GEODYN-II computations are presented in Table 3. The process of determining the orbits was performed separately for each orbital arc by the Bayesian least squares method using an iterative process (4–5 iterations per arc). The results of the Borowiec SLR station from July 1993 to December 2019 are given in 229 monthly arcs with a total of 18,397 and 12,354 normal points for LAGEOS-1 and LAGEOS-2, respectively. Several criteria used for rejecting normal points and orbital arcs are presented in Table 1. The percentage of rejected normal points for the laser station at Borowiec, after taking into account all the above criteria, was 7% and 6% for LAGEOSA-1 and LAGEOSA-2, respectively. This was mainly due to there being too small a number of normal points per month (below 50), creating the need to reject all points in such cases (56 months).

The quality of the orbits of the satellites was characterized by the dispersion of normal points in the form of the RMS of all stations with respect to the orbit. Figure 1 shows the RMS of the fit of all stations’ normal points for both satellites for each monthly arc. A significant improvement in the RMS can be seen in the first period of 1993–2002, the maintenance of the best results can be seen in the period 1998–2009, and a slight deterioration in the last period 2015–2019 resulting from the addition of several weaker stations as core stations for a better station distribution on the Earth’s surface can be seen.

## 4. Results-Position of Borowiec SLR Station

The results of the orbital analysis of the SLR station at Borowiec are presented in Table 4. For the evaluation of long-term changes, the entire observation period (1993–2019) was additionally divided into four sub-periods of several years: 1993–1997, 1998–2002, 2003–2009, and 2015–2019.

In order to assess the quality of determining station coordinates from a given arc, three parameters that determine the size of the standard deviation of the coordinates are important: the number of normal points, the RMS of the fit of normal points to the orbit (random deviations), and the long-term stability of deviations (systematic deviations). The number of normal points for both LAGEOS satellites is shown in Figure 2. It is noteworthy that the number of normal points in the studied period is very stable and the number of normal points is relatively low, with averages of 91 for LAGEOS-1 and 71 for LAGEOS-2, mainly due to the lack of daily observations and the bad weather in Borowiec (70% cloudiness).

The standard deviations of the coordinate determination are presented in Figure 3. The distribution of these results is similar to the distribution of normal points (Figure 2), which confirms the dependence of the standard deviation on the number of normal points.

The dependence of the standard deviation on the number of normal points per month is shown in Figure 4. There is a significant improvement in the standard deviation with an increase in the number of normal points. This figure confirms the correctness of the application of the criterion of a minimum of 50 normal points per month for rejecting monthly arcs.

The dispersion of normal points of a given station in relation to the orbit (RMS of fit) is another parameter that affects the precision of determining station coordinates and thus determines the accuracy of measurements. The results obtained for both LAGEOS satellites are shown separately in Figure 5. There is a clear improvement in the orbital RMS over time for both LAGEOS satellites.

Systematic deviations are characterized by range bias and its changes in the form of long-term stability. Long-term stability is defined as the standard deviation of monthly range bias estimates [1]. The range bias for both LAGEOS satellites should be kept constant so that its fluctuations do not significantly affect the quality of the orbit. The results are presented in Figure 6. In the period of study, all range biases were shifted by approximately −10 mm in comparison to ITRF2014. The long-term stability was significantly better in the period 2015–2019 (Figure 6 and Table 4).

The average RMS of the fit of the Borowiec SLR for the entire research period was 21.3 mm and 21.0 mm, the range bias was −7.3 mm and −6.3 mm, and the long-term bias stability was 11.0 mm and 12.4 mm for the LAGEOS-1 and LAGEOS-2 satellites, respectively (Table 4). This indicates a very good agreement of the results obtained independently for both satellites.

The positions of the stations in the form of geocentric coordinates X, Y, Z were determined independently for the epoch 2010.0 and on the first day of each month from each monthly orbital arc, for a total of 229 positions. All the determined station positions were transformed to the epoch 2010.0 by the station velocity from ITRF2014 (de-trended component). The geocentric positions obtained in this way are presented in Figure 7. The results show a significant jump in the Z component in 2002/2003. Arcs that contained less than 50 normal points for both the LAGEOS-1 + LAGEOS-2 satellites were rejected (55 arcs). Some arcs were removed due to them exceeding the 3xRMS deviations of individual components. The average geocentric coordinates for all periods and for the entire study are presented in Table 5.

Figure 7 and Table 5 show a clear 2 cm jump in the Z component in the period 2003–2009. To better assess the obtained results, the coordinates were transformed from the geocentric to the topocentric frame (length, width, height) [27]. Differences from the components of ITRF2014 allow for the assessment of changes in the horizontal and vertical planes. The results of the comparison of N, E, and U with the initial ITRF2014 coordinates are shown in Figure 8. These results confirm the jump in the vertical component, which indicates the effect of systematic apparatus error. This jump is also recorded in the ITRF2014 coordinates dated 27 July 2002 as two coordinate values before and after the jump. After introducing the coordinates of the new (after jump) value ITRF2014, the jump in the vertical component was doubled (Figure 9 top). This points to the existence of incorrect Borowiec coordinates in ITRF2014. After changing the coordinates according to the results in Table 5, the jump was completely removed (Figure 9 bottom). The final results of determining the topocentric coordinates after taking into account the above-mentioned correction are presented in Table 6.

The results obtained after the introduction of changes to ITRF2014 show a very good compliance of the period 2003–2009 compared to the other periods and a slight increase of several millimeters for the vertical component for the last period, 2015–2019.

The best estimation of the accuracy (difference between the true and computed values) of the station position is the repeatability or stability of the de-trended station coordinates X, Y, Z or N, E, U computed in the form of 3D RMS [28]. This value for the best stations reaches 4 mm. The stability results and standard deviations of the coordinates determined for annual periods are presented in Figure 10.

The best coordinate stability was obtained in the period 1998–2009, with the best result of 6.5 mm found in 2005. Similarly, for the standard deviations of the coordinate determination, the best result was 3.1 mm in 2004. The graphs in Figure 10 reflect the number of normal points presented in Figure 2; if there are more normal points, the accuracy is better. The stabilities of each component for Borowiec SLR for the whole period of study, 1993–2019, were equal to 13.2, 11.6, and 12.4 mm for X, Y, Z components, respectively, or 11.7, 12.2, and 13.2 mm for the N, E, U components. The 3D station coordinate stability was found to be 12.4 mm. The average 3D standard deviation of the determination of coordinates for both frames was equal to ±4.3 mm.

## 5. Results: Velocity of Borowiec SLR Station

The accurate determination of the velocity of the station movement is an important task for determining the position at any chosen time. The station velocity was determined by means of linear regression from the components of stations X, Y, Z or N, E, U determined for the first day of each month. The results are presented in Figure 11.

The length of the period from which the station velocity is determined has a very important role in determining the velocity of the station. This period should be several years, preferably more than 5 years; otherwise, the uncertainty in determining the velocity will be too great. The vertical velocity due to the movement of tectonic plates should be zero. The detailed results obtained for determining the velocity of the station at Borowiec are presented in Table 7. It is noteworthy that the 3D velocity determined in the entire period (24.9 mm/year) corresponds to the ITRF2014 value (25.0 mm/year) and, as expected, a low value for the vertical velocity (−1.8 mm/year). Similarly, for the X, Y, Z velocity components, we observed a very good agreement of the obtained results with ITRF2014.

An important parameter is the azimuth of station movement (*A*):(2)$$A=\tan^{-1}\frac{VE}{VN}$$
where *VE* and *VN* are *E* and *N* velocity components, respectively.

The azimuth for the entire period under study is 53.2° (the station moves in the northeast direction). The lower azimuth value in the period 2015–2019 (48.6°) may be the result of a shorter observation period of 4.4 years, which results in a lower accuracy in determining the velocity components *VN* and *VE*. All significant changes in *VN* and *VE* presented in Table 7, especially in the periods 1998–2002 (*VE*) and 2015–2019 (*VN*), can lead to significant biases in the azimuth.

## 6. Discussion and Conclusions

This paper presents a proposal for the assessment of the results of satellite laser ranging stations, which would allow for a detailed assessment of the station quality and its changes over time. The basic task was to precisely determine the coordinates of the stations and trace them throughout the period in which the laser observations of the satellites were carried out by the station.

In the case of the presented SLR station (7811) at Borowiec, we used the period from July 1993 to December 2019, with a few years’ break in observations in 2010–2014. The most significant change in the station position observed was a 2 cm jump in the vertical component, which was probably the result of the introduction of the new TENNELEC TC454 constant-fraction discriminator in early 2003, which lasted until the end of the Continuum laser’s operation—i.e., until 2010. This error could be due to the difference between the level of calibration and the level of observation. After a break in 2010–2014, due to the change in the characteristics of the laser pulse and the adjustment of the discriminator, this effect ceased to occur.

Taking into account the very long period of the study, it should be underlined that the results obtained in particular periods agree very well with the slight improvement in the recent period. The quality of the results is primarily decided by three factors: the number of normal points for each orbital arc; the dispersion of the normal points in relation to the orbit; and variations in systematic deviations—i.e., the long-term stability of range biases.

The number of normal points can be increased mainly by round-the-clock observations. Due to the low tracking accuracy of the SLR Borowiec telescope, daytime observations are currently not possible. The current telescope does not offer the possibility of significantly increasing the tracking accuracy below the 30” required for daytime observations. The only possibility is to construct a new telescope with much better parameters.

The second parameter limiting the quality of observations is the dispersion of the normal points in relation to the orbit. This parameter varied from over ±25 mm to ±18 mm in the recent period. For the best stations, this value reaches ±14 mm. In order to improve these random changes in the distance to the satellite, it is necessary to introduce new apparatus elements—in particular, an increase in the strength of the reflected signal from the satellite, an event timer, and a precisely calibrated constant-fraction discriminator are necessary. Obtaining strong reflected signals is important when using a photomultiplier tube; these can be obtained by reducing the laser beam divergence, which is in turn related to the tracking quality of the telescope.

Finally, the third parameter that decides the accuracy of a station is the variation in systematic deviations—i.e., the long-term stability of the range biases. This parameter for the SLR station at Borowiec remained at a level slightly above 10 mm in the first two periods, with a noticeable increase in the last two periods to 8 mm. This is still too great a value. The significant deterioration of these values was due to large variations in systematic deviations in the first two periods and a jump in the vertical component at the turn of 2002/2003. It is necessary to explain the inconsistency of the coordinate results with the ITRF2014 data, taking into account the approximately 2 cm jump in the vertical components on 27 July 2002.

The results of the SLR station at Borowiec to date should be assessed positively, taking into account the station’s equipment. However, the further improvement of the quality and quantity of results requires, above all, a new tracking telescope that would provide daytime observations as well as the possibility of operating a kilohertz laser, introducing SPAD, and reducing laser beam divergence. These activities, however, require very large financial outlays and take a long time to implement. It is possible to use the currently tested second telescope at Borowiec, provided that it meets the high requirements for tracking accuracy. Similar studies covering the presented area of analysis would allow for a more precise assessment of the operation of the SLR stations and enable us to find ways to significantly improve the SLR results.

## Figures and Tables

**Figure 1 sensors-22-00616-f001:**
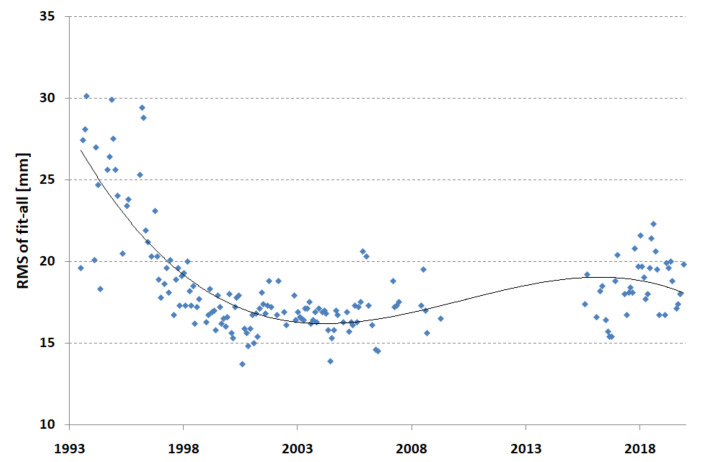
RMS of fit for all core stations (monthly points).

**Figure 2 sensors-22-00616-f002:**
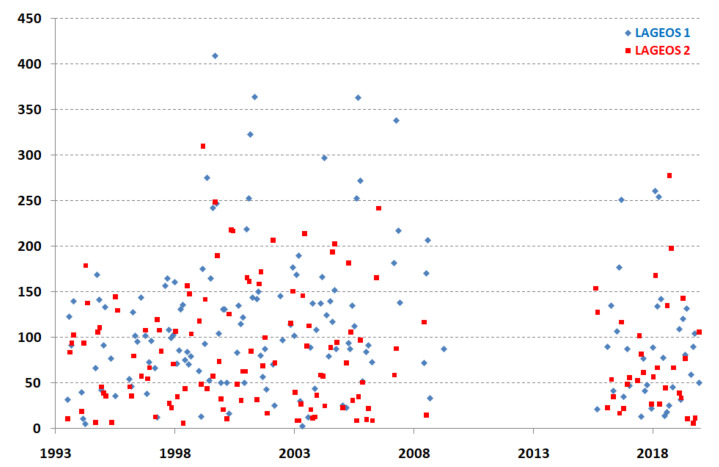
Borowiec SLR station—the number of normal points for the satellites LAGEOS-1 and LAGEOS-2 (monthly data).

**Figure 3 sensors-22-00616-f003:**
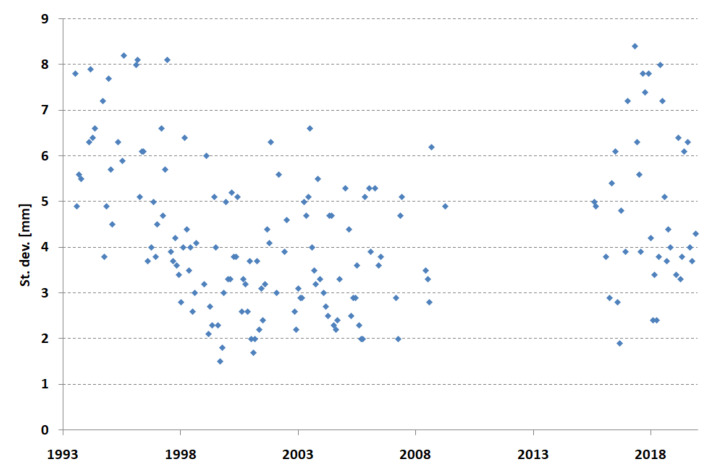
Borowiec SLR station—precision of the coordinate determination in 1993–2019.

**Figure 4 sensors-22-00616-f004:**
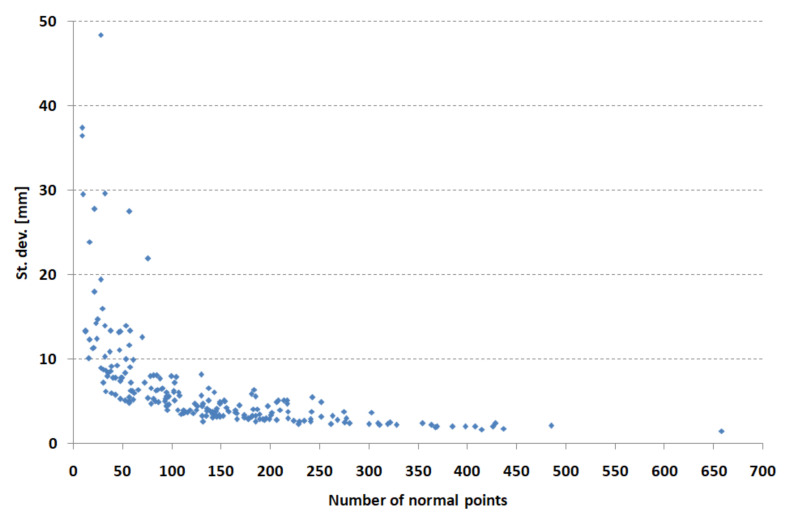
Dependence of the standard deviation of coordinate determination on the number of normal points for Borowiec SLR station.

**Figure 5 sensors-22-00616-f005:**
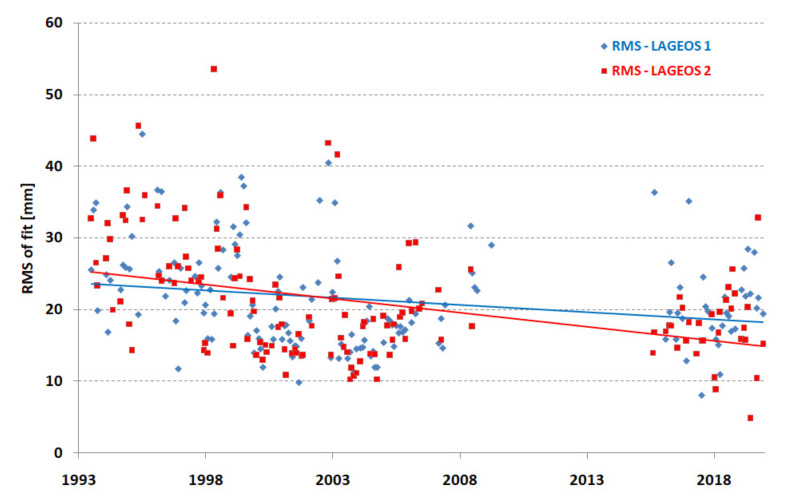
Borowiec SLR station—RMS of the fit of satellites LAGEOS-1 and LAGEOS-2 (monthly points).

**Figure 6 sensors-22-00616-f006:**
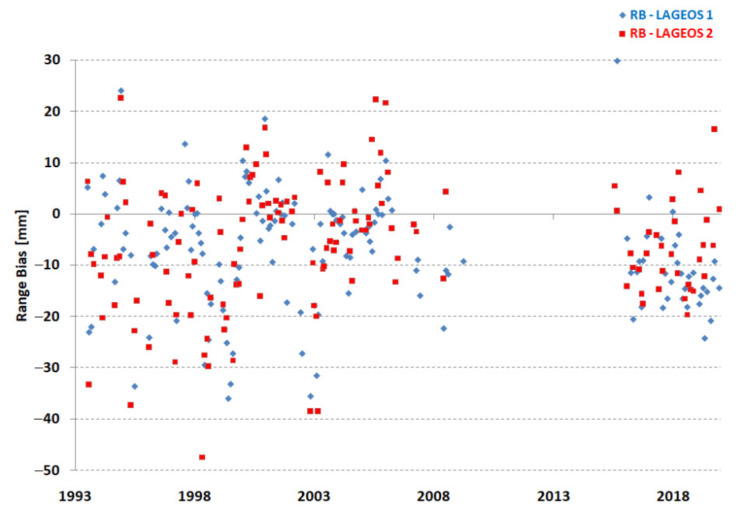
Borowiec SLR station—monthly range bias of satellites LAGEOS-1 and LAGEOS-2.

**Figure 7 sensors-22-00616-f007:**
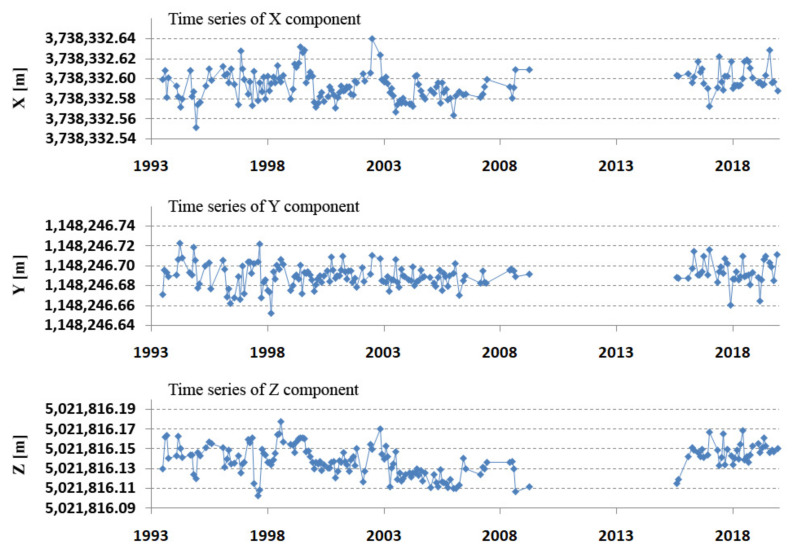
Borowiec SLR station—de-trended for epoch 2010.0 geocentric coordinates X, Y, Z (m) in 1993–2019.

**Figure 8 sensors-22-00616-f008:**
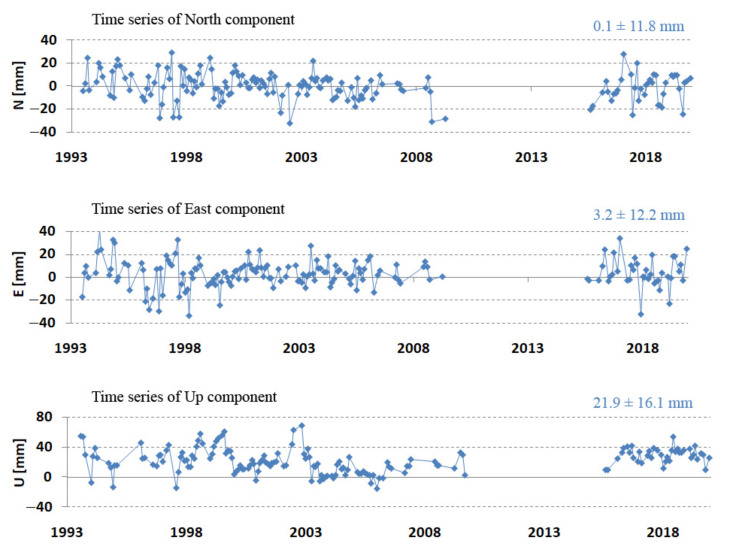
Borowiec SLR station—de-trended for epoch 2010.0 topocentric coordinates N, E, U (mm) in 1993–2019.

**Figure 9 sensors-22-00616-f009:**
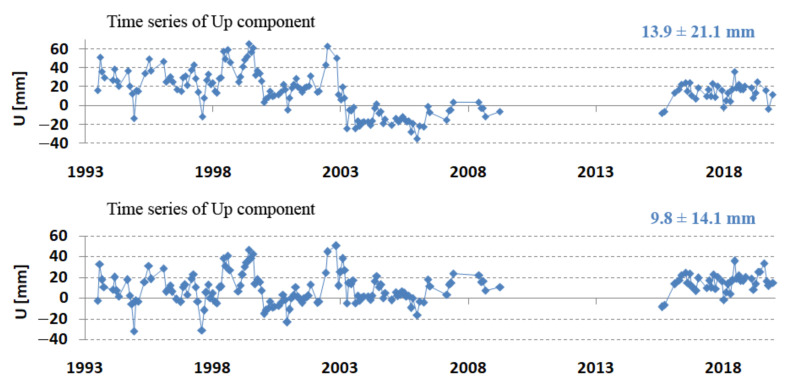
Borowiec SLR station—vertical component consistent with the ITRF2014 coordinate jump (**top**) and after the ITRF2014 data conversion (**bottom**).

**Figure 10 sensors-22-00616-f010:**
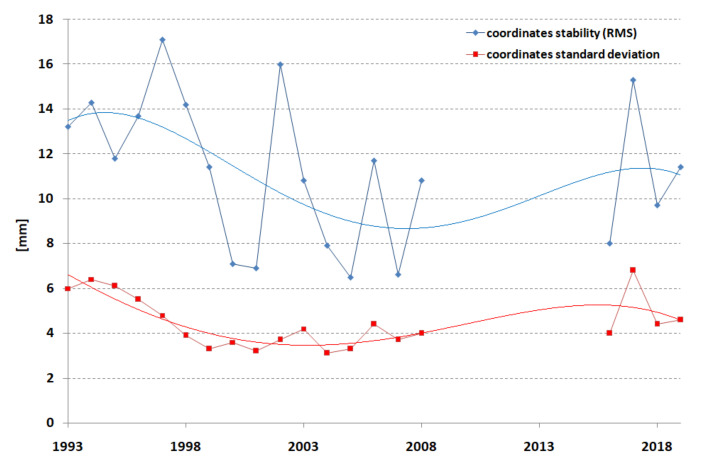
Borowiec SLR station—coordinate stability (blue) and coordinate standard deviations (red) for annual periods in 1993–2019.

**Figure 11 sensors-22-00616-f011:**
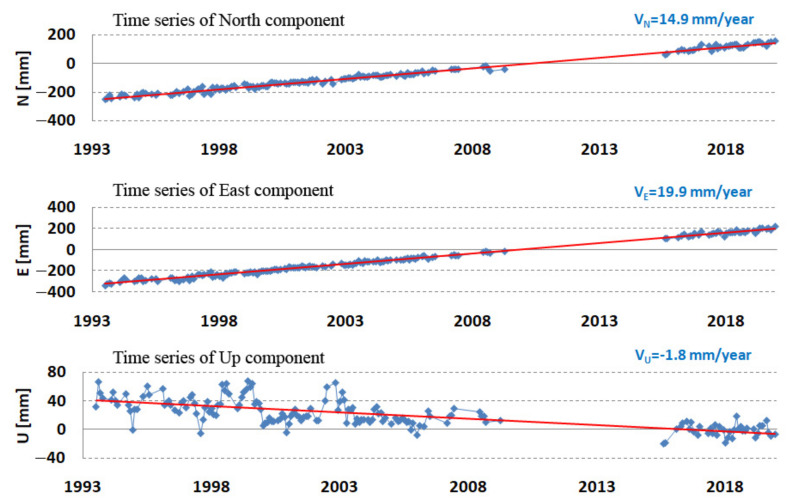
Borowiec SLR station—topocentric coordinates N, E, U (mm) in 1993–2019.

**Table 1 sensors-22-00616-t001:** History of the most important changes in the SLR system BORL7811 in the years 1993–2019 (YYYY—year; DDD—day of year).

YYYYDDD	SECTION	DESCRIPTION
1993202	TIME	Rb-frequency standard for PS-500 Timer
1994271	TIME	PS-500-2 Timer replaced PS-500 Timer
1995052	SOFTWARE	New controller computer PC-486, new real-time software, new tracking system
1995213	TIME	Rb-2 frequency standard replaced Rb-standard
1997034	ELECTRONICS	PMT HAMAMATSU H5023 replaced PMT RCA-8852
1997065	TELESCOPE	CCD camera installed
1997077	TIME	Cesium frequency standard replaced Rb-2 standard
1997110	ELECTRONICS	Amplifier HAMAMATSU C5594 installed
1997155	ELECTRONICS	PMT RCA-8852 replaced HAMAMATSU H5023
1997310	ELECTRONICS	PMT HAMAMATSU H5023 replaced RCA-8852
1998035	CALIBRATION	New ground target—distance 241.086 m
1998139	ELECTRONICS	Time Interval Counter STANFORD SR620 replaced PS-500-2
1998182	METEO	VAISALA meteo sensors installed; pressure PTB200A, temp. and humidity HMP45D
1998182	ELECTRONICS	Installation of the Time Interval Counter PS-500-2
2001342	TIME	New time and frequency source—cesium frequency standard HP5071A ver. 001
2001342	TELESCOPE	New CCD camera installed
2002127	ELECTRONICS	Time Interval Counter PS500-2 replaced by STANFORD SR620
2002319	ELECTRONICS	Time Interval Counter STANFORD SR620 replaced by STANFORD SR620B
2002323	ELECTRONICS	New start fast photodiode and discriminator TENNELEC 454 in start channel
2003088	ELECTRONICS	Discriminator B6 replaced by discriminator TENNELEC TC454 in stop channel
2007361	TELESCOPE	New cover of the main and secondary mirrors, prisms replaced by dielectric mirrors in Coude path, transmitting telescope (10 cm) installed with the system of the output beam divergence adjustment, air conditioning system for electronics and computers, new controller computers, new input/output software
2013352	LASER	Installation of an optical table for laser
2014021	LASER	Installation of a new laser EKSPLA PL-2250
2014203	TELESCOPE	A new optics was replaced in the receiving telescope including primary and secondary mirrors of the telescope
2014217	TELESCOPE	Milling of stands for dielectric mirrors of Coude path. Installation all mirrors (5 pieces) in Coude path
2014265	RECEIVER	Installation of a high-speed start Si photodiode FDS025 working in a range from 400 to 1100 nm
2014282	SOFTWARE	Delay changing in calibration program (from 50 to 10)
2015019	TELESCOPE	Two lenses mounting (on the telescope tube and movable lens inside the tube)
2015042	COMPUTER	New MASTER computer: Award Medallion BIOS v6.0, ASUS CUV4X ACPI BIOS Revision 1006, Intel (R) Pentium (R) III 1000 MHz, 768 MB
2015086	ELECTRONICS	Gate generator input set to 1.0 V, moment of the laser shot set to D gate generator output
2015126	ELECTRONICS	Time-lock in gate generator switch from T0 to CD (negated)
2016314	COMPUTER	New MASTER computer: Compaq, RAM 32 GB, Pentium Celeron 333 MHz, 4.7 GB 33 MHz, ISA 10 MB, PC ISA DB25
2017081	SYSTEM	New control unit responsible for steering of the laser beam during the observations
2017117	TELESCOPE	New mounting of the lens in the transmitter tube of the main telescope
2017145	RECEIVER	New start pulse for STANFORD counter from −0.4 V to −0.3 V
2017149	TELESCOPE	Observation filter change from 1 to 3
2017172	TELESCOPE	Stepper motors exchanged in the control unit responsible for steering of the laser beam during the observations
2017349	METEO	New pressure, temperature and humidity sensor has been installed (BOSCH 280)
2018037	RECEIVER	New amplitude of the start pulse for EKSPLA laser. New value is −2.5 V (100% of power)
2018046	TELESCOPE	Correction of the filter nr 7 (calibration tests)
2018068	TELESCOPE	Exchange of the green filter (Thorlabs FL532-10 CWL = 532 nm, FWHM = 10 nm)
2018078	RECEIVER	New trigger level for discriminator from −150 mV to −100 mV
2018149	RECEIVER	New settings of discriminator for starting photodiode (1.75 V)
2018187	TELESCOPE	New filters nr 4 and 6 (calibration tests)
2018200	RECEIVER	Power supply exchange for detector H5023 from HV to ZWN-24. The input voltage set to 2200 V
2019277	RECEIVER	The change of the “start level” of the discriminator from −180 mV to −500 mV, start pulse is −3 V for 100% of pulse energy

**Table 2 sensors-22-00616-t002:** Parameters of the Borowiec SLR station in 1993 and 2019.

Parameter	1993	2019
LASER		
Type Pulse energy (532 nm) Pulse width Repetition rate Divergence Energy stability	Nd:YAGCONTINUUM PY-62 25 mJ 100 ps 10 Hz 0.4 mrad 7%	Nd:YAGEKSPLA PL-2250 50 mJ 60 ps 10 Hz 0.4 mrad 0.5%
RECEIVING TELESCOPE		
Type Diameter of primary mirror Diameter of secondary mirror FOV Mount Tracking method Encoder resolution Tracking possibility	Cassegrain 65 cm 20 cm 5 arcmin Az-El Step by step 1.8 arcsec LEO, MEO	Cassegrain 65 cm 20 cm 5 arcmin Az-El Step by step 1.8 arcsec LEO, MEO
GUIDING TELESCOPE		
Type Diameter FOV Tracking control	Maksutov 20 cm 1 arcdeg visual	Maksutov 20 cm 1 arcdeg CCD camera
DETECTOR “START”	Avalanche photodiode	Si photodiode FDS025
DETECTOR “STOP” Quantum efficiency Gain TTS	PMT RCA-8852 10% 10^6^ 700 ps	PMT HAMAMATSU H5023 10% 10^6^ 160 ps
TIME INTERVAL COUNTER Accuracy Resolution	WUT PS-500-1 100 ps 20 ps	Stanford SR-620 25 ps 4 ps
TIME BASE	Cesium Frequency Standard Rhode and Schwarz	Active hydrogen maser CH1-75A
COMPUTER	MERA-400	PC-Pentium
CALIBRATION Type Target distance Method	PLATE External 1295.42 m PRE&POST	PLATE External 241.086 m PRE&POST
METEO Pressure Temperature Humidity	1 mbar 0.5° C 10%	BOSCH 280 0.1 mbar 0.2° C 3%
OPERATING STAFF	2 persons	1 person

**Table 3 sensors-22-00616-t003:** Force models and parameters for the GEODYN-II orbital software—LAGEOS satellites.

Force Models
Earth gravity field: EGM2008 20 × 20 [19]
Earth tides: IERS conventions 2003 [20]
Earth tide model: EGM96
Ocean tide model: GOT99.2 [21]
Third body gravity: moon, sun, and planets: DE403 [22]
Solar radiation pressure coefficient: C_R_ = 1.13
Tidal constants k_2_, k_3_, and phase k_2_: 0.3019, 0.093, 0.0 [23]
Earth albedo [10]
Dynamic polar motion [10]
Relativistic corrections [10]
**Constants**
Earth gravity parameter (GM): 3,986,004,415 × 10^14^ m^3^/s^2^
Speed of light: 299,792.458 km/s
Semi-major axis of the Earth: 6378.13630 km
Inverse of the Earth’s flattening: 298.25642
**Reference frame**
Inertial reference frame: J2000.0 Coordinates reference system: true of date at 0.0 h of the first day of the each month Station coordinates and station velocities: SLRF2014 for epoch 2010.0 [24]
Precession and nutation: IAU 2000
Polar motion: C04 IERS
Tidal uplift: model Love model H2 = 0.6078, L2 = 0.0847 [23]
Pole tide [10]
**Estimated parameters**
Satellite state vector (6 parameters)
Station geocentric coordinates (3 parameters)
Acceleration parameters: along track, cross track, and radial at 5-day intervals
**Measurement model**
Observations: 120-s normal points from EUROLAS Data Center
Laser pulse wavelength: 532 nm for all stations with the exception of 432 nm and 864 nm for stations 7810 (to 2008) and 7405 and 864 nm for station 7827
Centre of mass correction: 25.1 cm Cross-sectional area: 0.2827 m^2^ Mass of LAGEOS-1: 406.965 kg; mass of LAGEOS-2: 405.380 kg
Tropospheric refraction: model Mendes–Pavlis [25,26]
**Editing criteria**
All normal points > 5σ per arc Borowiec SLR station coordinates < 50 normal points per station per arc Borowiec SLR station coordinates > 2.5xsigma(3D) of position determination Borowiec SLR station coordinates > 3xRMS for each component north, east, and up
**Numerical integration**
Integration: Cowell’s method
Orbit integration step size: 120 s
Arc length: 1 month

**Table 4 sensors-22-00616-t004:** The results of the orbital analysis of the Borowiec SLR station (7811) data for LAGEOS-1 (L1) and LAGEOS-2 (L2) in 1993–2019.

Period	1993–2019	1993–1997	1998–2002	2003–2009	2015–2019
First data	July 1993	July 1993	January 1998	January 2003	August 2015
Last data	December 2019	December 1997	December 2002	April 2009	December 2019
Number of arcs	168	36	49	45	38
Number of NP all stations	1,758,469	295,632	491,024	573,876	397,937
RMS of fit (mm) all stations	18.5	22.7	16.9	16.8	18.6
Number of NP-L1 Borowiec	17,210	2785	6105	5291	3029
Number of NP-L2 Borowiec	11,662	2247	4175	2789	2451
RMS of fit-L1 (mm) Borowiec	21.3	25.6	21.9	18.3	20.3
RMS of fit-L2 (mm) Borowiec	21.0	28.1	20.5	18.6	17.4
Range bias-L1 (mm) Borowiec	−7.3	−4.9	−8.3	−5.0	−10.8
Range bias-L2 (mm) Borowiec	−6.3	−10.0	−7.0	−2.1	−6.9
Long-term bias stability-L1 (mm) Borowiec	11.0	11.6	13.0	8.6	9.4
Long-term bias stability-L2 (mm) Borowiec	12.4	13.0	14.7	11.5	8.4

**Table 5 sensors-22-00616-t005:** Average geocentric coordinates X, Y, Z of the Borowiec SLR station for each period.

Period	X [m]	Y [m]	Z [m]
1993–1997	3,738,332.5925 ± 0.0058	1,148,246.6915 ± 0.0058	5,021,816.1418 ± 0.0050
1998–2002	3,738,332.5968 ± 0.0035	1,148,246.6897 ± 0.0037	5,021,816.1432 ± 0.0030
2003–2009	3,738,332.5863 ± 0.0040	1,148,246.6877 ± 0.0037	5,021,816.1246 ± 0.0032
2015–2019	3,738,332.6011 ± 0.0051	1,148,246.6935 ± 0.0048	5,021,816.1459 ± 0.0044
**1993–2019**	**3,738,332.5940 ± 0.0045**	**1,148,246.6904 ± 0.0044**	**5,021,816.1385 ± 0.0038**

**Table 6 sensors-22-00616-t006:** Average topocentric coordinates N, E, U of Borowiec SLR station for each period after the ITRF2014 data conversion.

Period	N [mm]	E [mm]	U [mm]	3D st.dev. [mm]
1993–1997	2.9	4.4	6.7	±5.7
1998–2002	1.0	1.5	10.0	±3.5
2003–2009	−2.5	3.4	7.6	±3.7
2015–2019	−1.5	3.9	15.3	±4.9
**1993–2019**	**−0.1**	**3.2**	**9.8**	±4.3

**Table 7 sensors-22-00616-t007:** The velocities of Borowiec SLR station (7811) in 1993–2019.

	1993–2019	1993–1997	1998–2002	2003–2009	2015–2019	ITRF2014
Period (year)	26.5	4.5	5.0	6.3	4.4	
V_X_ component (mm/year)	−18.1 ± 0.3	−16.9 ± 1.7	−19.0 ± 1.1	−16.7 ± 1.3	−18.4 ± 1.6	−18.4
V_Y_ component (mm/year)	15.2 ± 0.2	13.4 ± 1.5	17.4 ± 0.8	15.8 ± 1.0	14.8 ± 1.4	15.1
V_Z_ component (mm/year)	7.7 ± 0.2	4.6 ± 1.5	4.6 ± 0.9	6.4 ± 1.1	11.0 ± 1.4	7.6
V_N_ component (mm/year)	14.9 ± 0.2	12.5 ± 1.5	13.2 ± 0.9	12.9 ± 1.1	17.2 ± 1.4	
V_E_ component (mm/year)	19.9 ± 0.2	17.7 ± 1.4	22.2 ± 0.8	20.0 ± 1.0	19.5 ± 1.3	
V_U_ component (mm/year)	−1.8 ± 0.3	−3.9 ± 1.6	−4.4 ± 1.0	−1.8 ± 1.2	0.6 ± 1.5	
3D velocity (mm/year)	24.9 ± 0.2	22.0 ± 2.7	26.2 ± 1.6	23.9 ± 2.0	26.0 ± 2.6	25.0
Horizontal velocity (mm/year)	24.9 ± 0.2	21.7 ± 1.9	25.8 ± 1.2	23.8 ± 1.4	26.0 ± 1.8	
Azimuth (°)	53.2 ± 0.9	54.8 ± 1.6	59.3 ± 1.0	57.2 ± 1.2	48.6 ± 1.5	

## Data Availability

Input SLR data is available at the open access EUROLAS Data Center (EDC).

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
