# Peer review of "Analysis of the Results of the Borowiec SLR Station (7811) for the Period 1993–2019 as an Example of the Quality Assessment of Satellite Laser Ranging Stations"

_sensors, 2022, doi:10.3390/s22020616_

Round 1

Reviewer 1 Report

Generally, nice, well-formulated paper.

I have only two small comments:

  1. Term "Range Bias" is written with capitals in this text. It is unclear why. Yes, it is a branche-specific term, but it is also e.g. "normal points". I suggest write it with small first letters, as it is already done in Table 4.
  2. Page 18, rows 328, 329. Yes, it is true in general, but the "sensitivity" is quantified in the last row of Table 7. It is not bad. The sentence should be re-formulated avoiding vague, non-technical formulations.

Author Response

Answers for reviewer #1 remarks (red)

Sensors, manuscript ID: sensors-1459491

Title: Analysis of the Results of the Borowiec SLR Station (7811) for the period 1993-2019 as an Example of the Quality Assessment of Satellite Laser Ranging Stations

Authors: S. Schillak, P. Lejba, P. Michałek, T. Suchodolski, A. Smagło, S. Zapaśnik

Generally, nice, well-formulated paper.

Thank you for your nice remark and a positive opinion.

I have only two small comments:

  1. Term "Range Bias" is written with capitals in this text. It is unclear why. Yes, it is a branche-specific term, but it is also e.g. "normal points". I suggest write it with small first letters, as it is already done in Table 4.

You are right that it is branche-specific term. We changed in the text all "Range Bias" to "range bias".  

  1. Page 18, rows 328, 329. Yes, it is true in general, but the "sensitivity" is quantified in the last row of Table 7. It is not bad. The sentence should be re-formulated avoiding vague, non-technical formulations.

Yes, it is not good form. New corrected sentence has been included in the text instead:

"All presented in Table 7 significant changes in VN and VE especially in the periods 1998-2002 (VE) and 2015-2019 (VN) lead to significant biases in azimuth."

Reviewer 2 Report

I think the paper should be extensively improved and enlarged ; in terms of SLR, stations, and orbit determination performances.

Please have a look to the recent paper (Earth, Planets and Space (Online) of Strugarek, Sosnica, Arnold, Jaggi and Zajdel (Determination of SLR station coordinates based on LEO, LARES, LAGEOS, etc.).

Best regards

Author Response

Answers for reviewer #2 remarks (red)

Sensors, manuscript ID: sensors-1459491

Title: Analysis of the Results of the Borowiec SLR Station (7811) for the period 1993-2019 as an Example of the Quality Assessment of Satellite Laser Ranging Stations

Authors: S. Schillak, P. Lejba, P. Michałek, T. Suchodolski, A. Smagło, S. Zapaśnik

I think the paper should be extensively improved and enlarged ; in terms of SLR, stations, and orbit determination performances.

Please have a look to the recent paper (Earth, Planets and Space (Online) of Strugarek, Sosnica, Arnold, Jaggi and Zajdel (Determination of SLR station coordinates based on LEO, LARES, LAGEOS, etc.).

Thank you for your remarks.

The main purpose of the presented article is to determine the coordinates of one satellite laser ranging station (SLR Borowiec - 7811) based on the results of all observations of SLR Borowiec performed with the third generation laser, i.e. from 1993 to 2019. Additionally, the quality of the results of these observations was assessed. The method currently used by the ILRS for determining the coordinates of the stations from the results of the LAGEOS-1 and LAGEOS-2 satellites observations was applied. Enlarged the scope of this article with additional SLR stations and new methods of the station coordinates determination from the results of the other SLR satellites significantly exceeds the assumed goal of this study.

Thank you for the example of the paper of Dariusz Strugarek et al. We know this work well, it was even presented by Dariusz Strugarek in October this year at our institute seminar. This is an excellent work to showcase the future possibilities of the stations coordinates determination from the results of satellite laser observations by using not only the results of the LAGEOS-1 and LAGEOS-2 satellites, but also the LARES satellite, eight LEO satellites equipped with GNSS receivers, thirteen Galileo satellites, and combined solutions. All of these methods performed inferior to the satellites LAGEOS-1 and LAGEOS-2, except for a weighted solution for all satellites. The paper covers a period of 1 year (2016), so obtained results should be treated as an introduction to further more detailed research on the multi-satellite method. The solution presented in the paper cannot be applied for such a long period of study (26.5 years), due to the lack of results in the 20th century and the next few years for LEO satellites equipped with GNSS receivers and GNSS satellites, as well as the absence of the LARES spherical geodetic satellite (launched in 2012). Only the LAGEOS-1 and LAGEOS-2 remained in the list of satellites. There is also no certainty that the results of the multi-satellite method will prove to be better than the solution used so far. In the opinion of the authors of this article, the improvement of the quality of determining the coordinates of stations from laser observations should be ensured by the third LAGEOS satellite, e.g. LARES-2, which will be launched soon, or the Russian glass satellite Geodetic Laser Autonomic Spherical Satellite (GLASS). Two-color SLR observations should provide a significant improvement in the quality of laser observations, but technical difficulties currently prevent the implementation of this method.

Added to the text in Introduction:"The multi-satellites method [12] based on LAGEOS-1, LAGEOS-2, LARES, Low Earth Orbit (LEO) satellites and Global Navigation Satellite Systems (GNSS) satellites is taken into account for determination the coordinates of the stations, but the results so far are not yet fully verified."

and in References:

  1. Strugarek D.; SoÅ›nica K.; Arnold D.; Jäggi A.; Zajdel R.; and Bury G. Determination of SLR station coordinates based on LEO, LARES, LAGEOS, and Galileo satellites. Earth Planets and Space 2021, 73(1), 87, 1-21. DOI: 10.1186/s40623-021-01397-1

Reviewer 3 Report

Manuscript Number: sensors-1459491

Full Title:  Analysis of the Results of the Borowiec SLR Station (7811) for the period 1993-2019 as an Example of the Quality Assessment of Satellite Laser Ranging Stations

The paper submitted to Sensors MDPI perform to investigate results of orbital analysis of the satellite laser ranging data performed by the Borowiec SLR station (Poland) in the period from July 1993 to December 2019, including the determination the station positions and velocity. The topic is of considerable interest to the scientific community and the paper is analysed in original submission.

The work is pleasurable to read, also because it is written in excellent English language. The paper has been edited according to the Sensors MDPI instructions for authors and is set up correctly in all its paragraphs. In my opinion, the study is adequately original and the methodological approach is rigorous. The introduction synthesizes the scientific background. The methods and tools are described with details to allow another researcher to reproduce the results. The results are interpreted in appropriate way and the data are robust (26 years) enough to draw the conclusions.

On the basis of what has been mentioned, I may propose to agree in the present form, but there is only few criticism, in my opinion regarding some figures in the text.

  • Line 160-178, please modify table 3 because line numbers are inserted inside the table and are not readable by the reader;
  • Line 255, figure 7, please modify figure 7, because it is of low quality and too small;
  • Line 273, figure 8, please modify figure 8, because it is of low quality and too small;
  • Line 277, figure 9, please modify figure 9, because it is of low quality and too small;
  • Line 309, figure 11, please modify figure 11, because it is of low quality and too small;
  • Line 403, please chech the URL because is Page Not Found on ilrs.gsfc.nasa.gov.

In conclusion the work deserves publication after minor revisions, in my opinion.

Best regards

Author Response

Answers for reviewer #3 remarks (red)

Sensors, manuscript ID: sensors-1459491

Title: Analysis of the Results of the Borowiec SLR Station (7811) for the period 1993-2019 as an Example of the Quality Assessment of Satellite Laser Ranging Stations

Authors: S. Schillak, P. Lejba, P. Michałek, T. Suchodolski, A. Smagło, S. Zapaśnik

The paper submitted to Sensors MDPI perform to investigate results of orbital analysis of the satellite laser ranging data performed by the Borowiec SLR station (Poland) in the period from July 1993 to December 2019, including the determination the station positions and velocity. The topic is of considerable interest to the scientific community and the paper is analysed in original submission.

The work is pleasurable to read, also because it is written in excellent English language. The paper has been edited according to the Sensors MDPI instructions for authors and is set up correctly in all its paragraphs. In my opinion, the study is adequately original and the methodological approach is rigorous. The introduction synthesizes the scientific background. The methods and tools are described with details to allow another researcher to reproduce the results. The results are interpreted in appropriate way and the data are robust (26 years) enough to draw the conclusions.

Thank you for your nice remarks and a positive opinion.

On the basis of what has been mentioned, I may propose to agree in the present form, but there is only few criticism, in my opinion regarding some figures in the text.

  • Line 160-178, please modify table 3 because line numbers are inserted inside the table and are not readable by the reader;

Table 3 has been modified.

  • Line 255, figure 7, please modify figure 7, because it is of low quality and too small;
  • Line 273, figure 8, please modify figure 8, because it is of low quality and too small;
  • Line 277, figure 9, please modify figure 9, because it is of low quality and too small;
  • Line 309, figure 11, please modify figure 11, because it is of low quality and too small;

All figures have been modified.

  • Line 403, please check the URL because is Page Not Found on ilrs.gsfc.nasa.gov.

Sorry, these are NASA limitations. You need a login and password. You can try to enter just ilrs.

In conclusion the work deserves publication after minor revisions, in my opinion.

Thank you and best regards